# Design and Application of Memristive Balanced Ternary Univariate Logic Circuit

**DOI:** 10.3390/mi14101895

**Published:** 2023-09-30

**Authors:** Xiaoyuan Wang, Xinrui Zhang, Chuantao Dong, Shimul Kanti Nath, Herbert Ho-Ching Iu

**Affiliations:** 1Wenzhou Institute of Hangzhou Dianzi University, Wenzhou 325024, China; 2School of Electronics and Information, Hangzhou Dianzi University, Hangzhou 310018, China; z120431540@163.com (X.Z.); crazypink@163.com (C.D.); 3School of Photovoltaic and Renewable Energy Engineering, University of New South Wales (UNSW Sydney), Kensington, NSW 2052, Australia; shimul_kanti.nath@unsw.edu.au; 4School of Engineering, The University of Western Australia, Crawley, WA 6009, Australia; herbert.iu@uwa.edu.au

**Keywords:** memristor, balanced ternary, univariate logic, combinational logic circuit

## Abstract

This paper proposes a unique memristor-based design scheme for a balanced ternary digital logic circuit. First, a design method of a single-variable logic function circuit is proposed. Then, by combining with a balanced ternary multiplexer, some common application-type combinational logic circuits are proposed, including a balanced ternary half adder, multiplier and numerical comparator. The above circuits are all simulated and verified in LTSpice, which demonstrate the feasibility of the proposed scheme.

## 1. Introduction

In the era of big data, the amount of data is growing explosively, and as a result, digital logic systems are having difficulty in processing such huge amounts of data while striving for ever-increasing efficiency [1]. To meet the demand of data processing speed and power efficiency, ternary logic has received recent attention due to its advantages of higher single-line information carrying capacity and additional logical functions [2,3,4,5,6,7]. Compared to the binary digital signal, each bit of the ternary digital signal contains more information, resulting in a higher transmission rate at the same frequency. It also helps in reducing circuit interconnections, and digital chips can be made smaller and less expensive [8,9,10]. Ternary logic can be divided into two categories: balanced ternary {−1, 0, 1} and unbalanced ternary {0, 1, 2} or {0, −1, −2} [11]. Among them, balanced ternary logic has unique advantages, including the ability of having a unified representation for positive and negative numbers without the sign bit, and multiplication operation without generating a carry. Moreover, the symmetry of one-bit addition and multiplication operations can be used for symmetric arithmetic operation circuit design [12,13].

In recent years, ternary digital logic circuits have been implemented in various technologies, including MOSFETs, carbon nanotube field effect transistors (CNTFETs), resonant tunneling diodes (RTD), single-electron transistors, memristors, etc. [14,15,16,17,18]. Among them, memristor-based ternary logic is of considerable interest, as it provides the advantages of non-volatility, nanoscale and compatibility with CMOS technology [19,20].

There are two typical paradigms for designing memristor-based ternary logic circuits; one uses three resistance states of the ternary memristor and the other one uses the voltage value as the logic variable, where the former method makes full use of the resistance change characteristics of the memristor, the operation result can be stored in memristors, the logic state will not be lost after power withdrawal. Several studies [21,22] reported on the unbalanced ternary basic logic gate circuit using the three resistance states of the ternary memristor which correspond to positive ternary logic ‘0’, ‘1’, and ‘2’. A voltage-controlled tri-valued memristor model was first proposed in Ref. [22], with designs of ternary AND, OR and NOT gate circuits based on it. In this case, three stable resistance states, *R*_H_, *R*_M_ and *R*_L_, correspond to logic ‘0’, ‘1’, and ‘2’, respectively. In Ref. [23], a bipolar three-state ZnO memristor was reported, and then all the 27 possible univariate positive ternary logics were realized with a single memristor cell. Furthermore, Ref. [24] proposed a method of realizing a balanced ternary adder using the resistance state transformation of only one single memristor, in which the circuit area and system power consumption were greatly reduced.

Significant advancement has also been achieved in implementing logic circuits using the second method (i.e., employing the voltage value as a logic variable) [25,26]. For example, Wang et al. [27] reported the construction of positive ternary logic circuits, including, the ternary AND gate, OR gate, inverters, encoder and decoder circuits. Similarly, in Ref. [28], ternary basic logic gates and combinational logic circuits using memristor-CNTFET hybrid circuit were proposed, whose delay and circuit complexity were lower compared to those of the circuits only using CNTFETs. Ref. [29] proposed a systematic method of constructing a two-digit ternary logic function based on the concept of memristive threshold logic (MTL) and applied this method for constructing basic ternary arithmetic operations. Compared to that of the previously reported relevant circuit design schemes, the circuit area of the ternary adder and ternary multiplier was greatly reduced. In Refs. [5,30], the balanced ternary logic circuits based on a memristor and MOSFET were proposed. The design idea was to construct balanced ternary essential logic gates, such as TAND, TOR, TI, TSUM, NCONS, NANY, etc., and then propose design scheme of a balanced ternary full adder.

As a further development in the present study, combinatorial logic circuits are implemented directly by combining univariate logic circuits and multiplexers. The multiplexer uses the circuit proposed in Ref. [31], and its function is to select only one of the data of multiple channels and transmit it to the output terminal according to the state of the selection signal. The proposed design scheme of a memristive balanced ternary digital logic circuit with the voltage value as the logic variable could be beneficial for further improving information storage, processing, and transmission efficiency.

The structure of this paper is as follows: Section 2 presents a design scheme of a balanced ternary single-variable logic function circuit based on a hybrid design of memristor and MOS transistor; in Section 3, based on the proposed univariate logic circuits and the multiplexer designed in our previous study [31], balanced ternary application-type combinational logic circuits are designed, including a half adder, multiplier, and numerical comparator; Section 4 presents the comparison and analysis of the proposed circuit with existing designs; Section 5 contains the conclusion of this paper.

## 2. Balanced Ternary Univariate Logic Circuit

In digital logic circuits, univariate logic functions are used to perform corresponding logic transformations on signals, thus playing an important role in circuit design. For ternary logic, there are three possible values for a single-input variable, with 3^3^ = 27 possible output results in total, as shown in Table 1.

As evident from Table 1, balanced ternary univariate logic can be divided into three categories, such as three-state to one-state logic, three-state to two-state logic, and three-state to three-state logic. The first category (three-state to one-state logic: *F*_1_, *F*_14_, and *F*_27_) is also called constant logic; that is, irrespective of the input value, the output is a fixed logic state, and therefore their applications are limited in circuit design.

This paper will mainly involve the circuit design of the other two categories, i.e., the balanced ternary three-state to two-state logic, and the three-state to three-state univariate logic, as well as a detailed analysis and simulation verification of the corresponding circuits. All univariate logic circuits are represented by the circuit symbol shown in Figure 1.

### 2.1. Three-State to Two-State Logic

From the truth table of the balanced ternary univariate logic function shown in Table 1, there are 18 kinds of univariate logic functions for three-state to two-state logic. Among them, the logics of *F*_19_ and *F*_25_ correspond to the NTI gate and the PTI gate, respectively, which have been introduced in detail in Ref. [31] and will not be repeated in this section.

#### 2.1.1. Circuit Design of Logic Function *F*_4_, *F*_5_, *F*_9_, *F*_10_, *F*_13_, *F*_18_, *F*_23_ and *F*_26_

Table 2 shows the designed circuit diagram and the threshold voltage range of the MOS transistor. While the circuits of logic functions *F*_4_ and *F*_9_ only need one memristor and one NMOS transistor, those of the logic functions *F*_5_, *F*_9_, *F*_10_, *F*_13_, *F*_18_, *F*_23_ and *F*_26_ are all composed of two memristors and one NMOS transistor. Among them, two groups of logic (*F*_10_ and *F*_13_) and (*F*_23_ and *F*_26_) adopt the same circuit structure, but the difference is that the threshold voltage ranges of MOS transistors in the corresponding circuits are different. See Table 2 for details.

The working principles of these logic functions can be understood via simply analyzing the circuits of logic functions *F*_4_ and *F*_5_. For *F*_4_, when input A is −V_DD_ (logic ‘−1’) or 0V (logic ‘0’), transistor *T*_1_ is turned off, and the output terminal will be directly connected to the input terminal through memristor *M*_1_, so the output remains consistent with the input. When input A is V_DD_ (logic ‘1’), transistor *T*_1_ is turned on, and the output terminal will be directly connected to -V_DD_ through *T*_1_, that is, logic ‘-1’ is the output. For *F*_5_, when input A is −V_DD_ (logic ‘−1’) or 0V (logic ‘0’), transistor *T*_1_ is turned off, the output terminal will pass through memristor *M*_1_, which is directly connected to the input terminal, and the output is consistent with the input. When input A is V_DD_ (logic ‘1’), transistor *T*_1_ is turned on, and there is a current path flowing from the input terminal to −V_DD_ in the circuit. Both memristors *M*_1_ and *M*_2_ are switched to the *R*_OFF_ state, and the output terminal is about 0 V after voltage division, that is, the output logic is ‘0’. Similar methods can be used to verify the correctness of other circuits, which will not be repeated here.

#### 2.1.2. The Circuit Design of the Remaining Three-State to Two-State Logic Function

The remaining three-state to two-state logic function circuits, including *F*_2_, *F_3_*, *F*_7_, *F*_11_, *F*_15_, *F*_17_, *F*_21_, and *F*_24_ logic, can be obtained via cascading the circuits as mentioned above. For example, for the *F*_2_ logic circuit, it is only necessary to cascade an *F*_4_ logic circuit after the *F*_26_ logic circuit to complete the logic conversion corresponding to *F*_2_. As shown in Table 3, it is a design scheme of a single-variable three-state to two-state logic circuit designed via the cascade method. Among them, ‘*F*_m_ + *F*_n_’ indicates that the *F*_n_ logic circuit is cascaded after the *F*_m_ logic circuit.

#### 2.1.3. Simulation Verification of Three-State to Two-State Logic Circuit

To validate the above approach, the proposed circuit is simulated and verified using LTSpice. Figure 2, Figure 3 and Figure 4 show the simulation waveforms of three kinds of three-state to two-state logic, including the transition from the three-state logic circuits to logic (−1,1), (−1,0) and (0,1).

### 2.2. Three-State to Three-State Logic

There are 6 types of single-variable logic functions in this category, including *F*_6_, *F*_8_, *F*_12_, *F*_16_, *F*_20_ and *F*_22_. Among them, the output of *F*_6_ is equal to the input, which is called ‘follower logic’. Only five types of three-state to three-state logic are effective and used in circuit design. However, the *F*_22_ logic (STI gate) circuit has been discussed in detail previously [31], and the remaining four logic circuits will be introduced here.

#### 2.2.1. Circuit Design of Up-Spin Logic Function *F*_16_ and Down-Spin Logic Function *F*_20_

The circuit structure diagram of the up-spin logic function, *F*_16_, the down-spin logic function, *F*_20_, and the threshold voltage range of the MOS transistor used is shown in Table 4. The *F*_16_ circuit uses two memristors and two NMOS transistors, while the *F*_20_ circuit uses three memristors and three NMOS transistors. In the case of *F*_20_, when input A is −V_DD_ (logic ‘−1’), MOS transistors *T*_1_, *T*_2_, and *T*_3_ are all turned off, and the output terminal is pulled up to V_DD_ through memristor *M*_1_, that is, the output logic is ‘1’. When input A is 0 V (logic ‘0’), both *T*_1_ and *T*_2_ are turned off, *T*_3_ is turned on, and the output terminal is directly connected to −V_DD_ through *T*_2_, that is, the output logic is ‘−1’. When input A is V_DD_ (logic ‘1’), both *T*_1_ and *T*_2_ are turned on, *T*_3_ is turned off, and there is a current path from V_DD_ to −V_DD_ in the circuit. Both memristors *M*_1_ and *M*_2_ are switched to the *R*_OFF_ state, and the output terminal outputs a voltage nearly 0V, that is, the output logic is ‘0’. The correctness of spin-up logic function *F*_16_ can be verified via a similar method, which will not be repeated here.

#### 2.2.2. The Circuit Design of the Remaining Three-State to Three-State Logic Function

The remaining three-state to three-state logic function circuits, including *F*_8_ and *F*_12_ logic, can also be obtained via cascading the circuits mentioned above. For example, for the *F*_8_ logic circuit, it is only necessary to cascade an *F*_22_ logic circuit after the *F*_20_ logic circuit to complete logic conversion corresponding to *F*_8_. Similarly, the *F*_12_ logic circuit can be obtained via cascading *F*_16_ logic and *F*_22_ logic. Table 5 shows the design scheme of the univariate three-state to three-state logic circuit using the cascade method. The term ‘*F*_m_ + *F*_n_’ indicates that the *F*_n_ logic circuit is cascaded after the *F*_m_ logic circuit.

#### 2.2.3. Verification of Three-State to Three-State Logic Circuit Using LTSpice Simulation

The above circuit was simulated in LTSpice, which provides a verification of the design for a given input signal. The simulation waveform diagram of the three-state to three-state logic circuit is shown in Figure 5.

## 3. Design of Balanced Three-Valued Combinational Logic Circuit Based on Univariate Logic and Multiplexer

A multiplexer can select one of several input signals to the output. This paper uses the balanced ternary multiplexer circuit proposed in Ref. [31], which can realize the output of one signal from the three inputs. The corresponding input–output relationship is expressed as follows:(1)$$Y=\left\{ \begin{array}{ll} I_{-1} & S=-1\\ I_{0} & S=0\\ I_{1} & S=1 \end{array}\right.$$

Here, *S* is a selection signal, and I_−1_, I_0_, and I_1_ are three input signals. The multiplexer is composed of a balanced ternary one-line–one-line decoder, three balanced ternary minimum gates and one balanced ternary maximum gate. The circuit structure diagram is shown in Figure 6.

When the selection signal is *S =*−1, the output terminals *S*_−1_, *S*_0_ and *S*_1_ of the one-line—three-line decoder output logic 1, −1 and −1, respectively. According to the working principle of the minimum value gate and the maximum value gate, the output signal of the circuit is equal to input signal I_−1_, that is, *Y* = I_−1_, and the circuit realizes the function of output signal I_-1_. When the selection signal *S* = 0, the output terminals *S*_−1_, *S*_0_ and *S*_1_ of the decoder output the logic −1, 1, −1, respectively. In this case, *Y* = I_0_, that is, the circuit realizes the function of outputting signal I_0_. Finally, when the selection signal *S* = 1 occurs, the output o decoder terminals *S*_−1_, *S*_0_, and *S*_1_ output the logic −1, −1, and 1, respectively, resulting in *Y* = I_1_.

In this paper, a balanced ternary half adder, a balanced ternary multiplier and a balanced ternary numerical comparator are also designed using the multiplexer and the univariate logic circuit described in Section 2. The truth tables and circuit structures of these applications are summarized in Table 6 and Table 7, respectively. The design process and working principle of each circuit are explained in the following three subsections, along with the corresponding simulation results.

### 3.1. Balanced Ternary Half Adder

It can be seen from the truth table that when input signals *A* = −1 and *B* selects f the values of {−1, 0, 1}, the ‘SUM’ outputs the sum of the half adder outputs, corresponding to {1, −1, 0}. According to the working principle of the multiplexer, if *A* is used as the selection signal, we can obtain the following results. When *A* = −1, the multiplexer selects the I_-1_ input terminal for the output, that is, SUM = I_-1_. And as shown in Table 1 a univariate logic *F*_20_ just can fulfill the conversion demanded in the red square in Table 6, so *F*_20_ is selected to connect the input B and I_-1_ in the circuit. Similarly, when input signal *A* = 0, the sum output of the half adder is SUM = I_0_ = *B*, so we directly connect *B* to I_0_. When input signal *A* = 1, SUM = I_1_, the logic *F*_16_ is consistent with the conversion, so *F*_16_ is selected to connect the input *B* and I_1_ in this case. The ‘CARRY’ output circuit part is designed in the same way. Figure 7 shows the corresponding logic conversion diagram of the balanced ternary half adder.

According to the univariate logic function relationship in Table 1, for the ‘sum’ output part, the three logic conversion relationships correspond to the down-spin logic function, *F*_20_, the follow-up logic function, *F*_6_, and the up-spin logic function, *F*_16_. For the ‘carry’ output part, the three logical conversion relationships correspond to the logical functions *F*_5_, *F*_14_, and *F*_15_. Therefore, it is only necessary to introduce the corresponding univariate logic circuit into circuit design. The LTSpice simulation waveform diagram is given in Figure 8.

### 3.2. Balanced Ternary Multiplier

Balanced ternary does not generate carry during multiplication, so it has certain advantages over the unbalanced ternary logic. The multiplier circuits design is as follows: When *A* = −1, the multiplexer selects the I_-1_ input terminal for the output, According to Table 1 and Table 6, *F*_22_ can be selected to connect the input *B* and I_-1_ in the circuit. When *A* = 0, the I_0_ terminal of the multiplexer is gated, and now the output terminal outputs a logic ‘0’, so we can directly connect I_0_ to the ground. When *A* = 1, the logic value of the output terminal is consistent with the input signal *B*, so we connect input signal *B* to the I_1_ terminal of the multiplexer in this case. Figure 9 shows the LTSpice simulation waveform diagram of the circuit.

### 3.3. Balanced Ternary Numerical Comparator

As we known, the output of multiplexer equals to I_-1_ when the input signal *A* is selected as −1, that is, MLE = I_-1_. And according to the truth Table 1 and Table 6, logic *F*_10_ performs the some function when input *A* = −1. so *F*_10_ is selected to connect the input *B* and I_-1_ in the circuit. Similarly, logics *F*_22_ and *F*_26_ are chosen to perform the corresponding functions when input *A* = 0 and *A* = 1. Figure 10 shows the simulation results for a balanced ternary numerical comparator.

## 4. Comparison and Analysis

The number of components using the proposed method are given in Table 8 and are compared with that reported earlier [31]. It is evident that there are significant advantages of the proposed method in terms of the balanced ternary half adder, multiplier, and numerical comparator circuit as the number of circuit components is reduced by 37.8%, 39.5%, and 48.2%, respectively.

A comparison of power consumptions of the circuits in Ref. [31] is also given in Table 9, including static power consumption, average power consumption and dynamic power consumption. The static power consumption shown in this table is the maximum static power consumption value of all the nine input combinations, and the average power consumption is the average static power consumption for every input combination. The dynamic power dissipation was estimated and calculated in accordance with the following formula:*P*(dynamic) = |*P*(max) − *P*(avg)| (2)
where *P*(max) means the instantaneous maximum power consumption, and the *P*(avg) is the average power dissipation, which can be obtained through SPICE simulations.

It is evident that, there are no significant advantages over the method reported in Ref. [31] for the balanced ternary half adder, multiplier and numerical comparator circuits. Particularly, THA’s static power consumption exceeds about three times that in Ref. [31]. This is because we use three more transistors with relatively higher power consumption than that in Ref. [31]. However, our dynamic power consumption is relatively lower for our present study, showing the significance of the current approach.

## 5. Conclusions

In summary, a design scheme for a balanced ternary logic circuit based on a memristor and MOS transistor was proposed. At first, the design of a balanced ternary single-variable logic circuit was introduced, including the commonly used three-state to two-state logic and three-state to three-state logic. Then, combined with the balanced ternary multiplexer, several design schemes of application-type combinational logic circuits were proposed, including a balanced ternary half adder, multiplier and numerical comparator. The designed circuits were simulated and further verified using LTSpice. Finally, the proposed circuit was compared with other design methods. Our results show that the number of components can be significantly reduced using the proposed design method, which could further reduce the complexity of the circuit.

## Figures and Tables

**Figure 1 micromachines-14-01895-f001:**
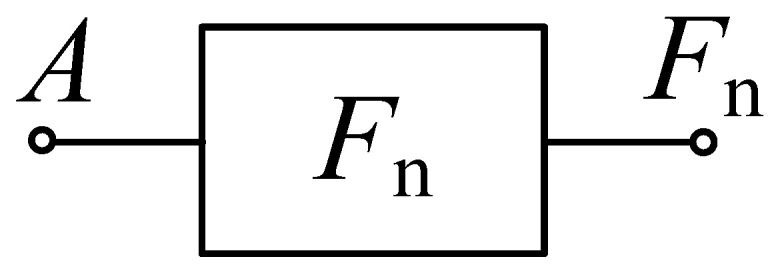
Circuit symbol of univariate logistic function *F*_n._

**Figure 2 micromachines-14-01895-f002:**
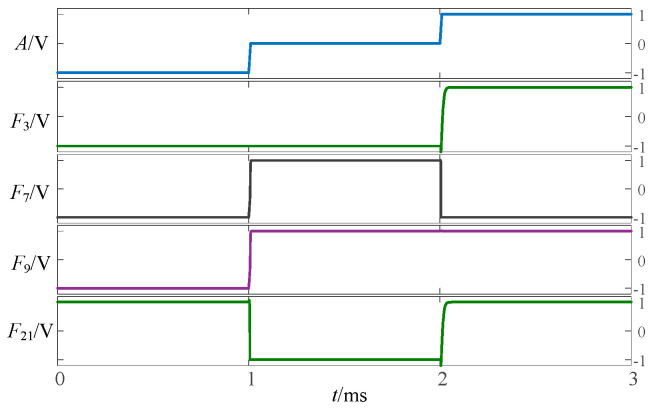
Simulation results of transition from three-state logic circuits to logic (−1,1).

**Figure 3 micromachines-14-01895-f003:**
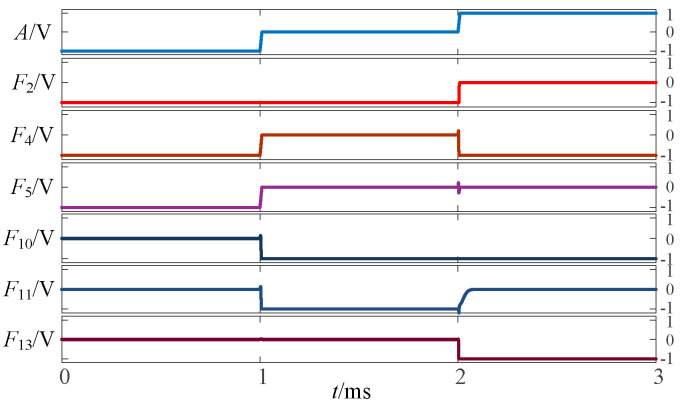
Simulation results of transition from three-state logic circuits to logic (−1,0).

**Figure 4 micromachines-14-01895-f004:**
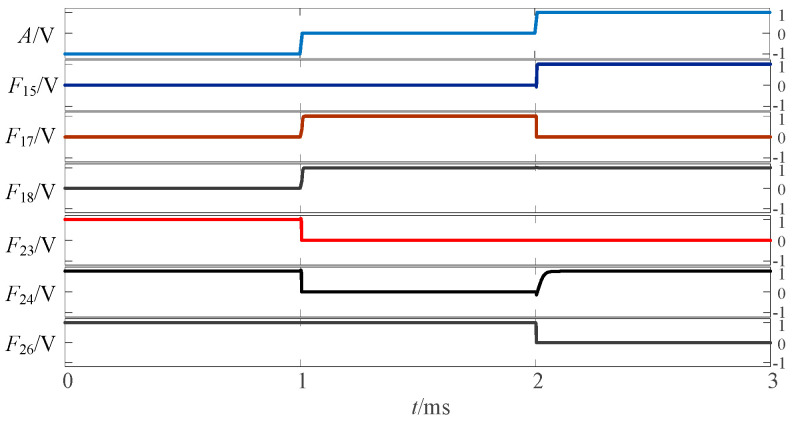
Simulation results of transition from three-state logic circuits to logic (0,1).

**Figure 5 micromachines-14-01895-f005:**
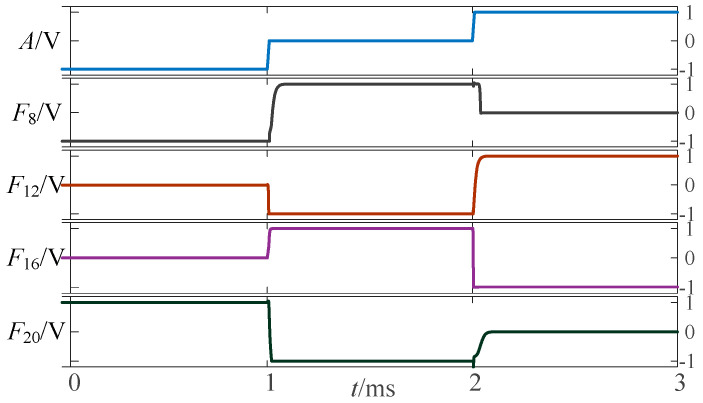
Simulation waveform diagram of three-state to three-state univariate logic circuit.

**Figure 6 micromachines-14-01895-f006:**
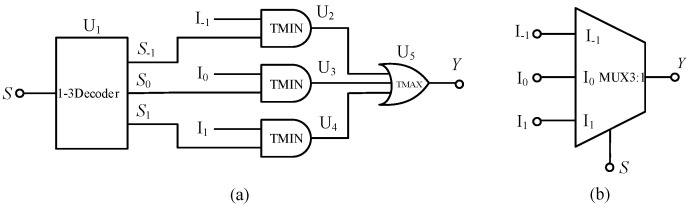
(**a**) Circuit diagram of balanced ternary multiplexer (**b**) symbol.

**Figure 7 micromachines-14-01895-f007:**
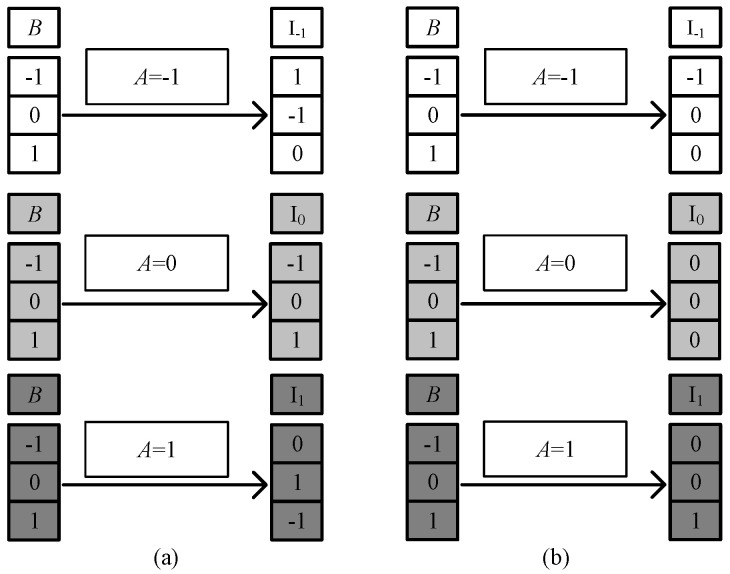
The corresponding logic conversion diagram of the balanced ternary half adder. (**a**) ‘sum’ output part; (**b**) ‘carry’ output part.

**Figure 8 micromachines-14-01895-f008:**
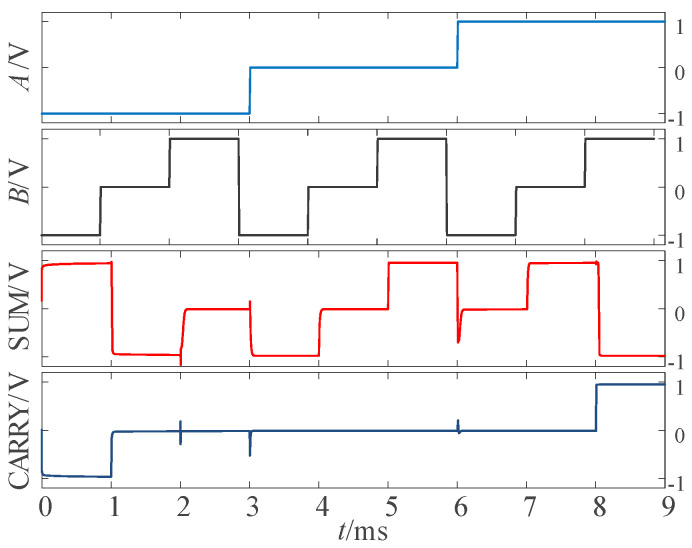
Simulation waveform diagram of half adder.

**Figure 9 micromachines-14-01895-f009:**
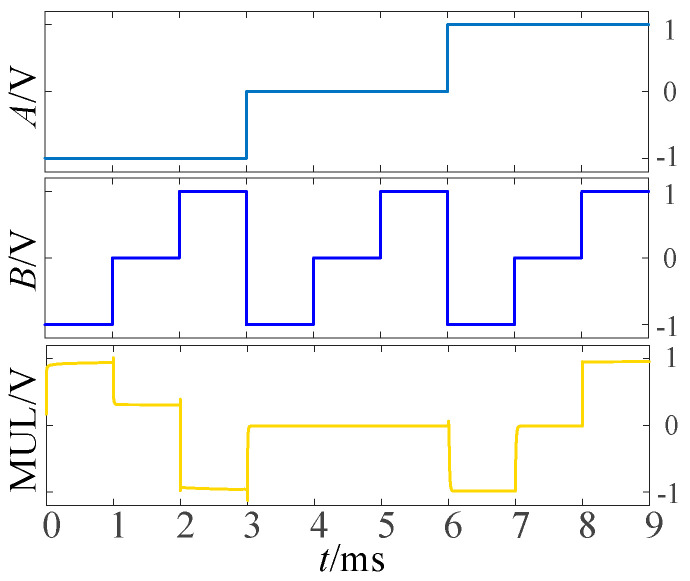
Simulation diagram of multiplier.

**Figure 10 micromachines-14-01895-f010:**
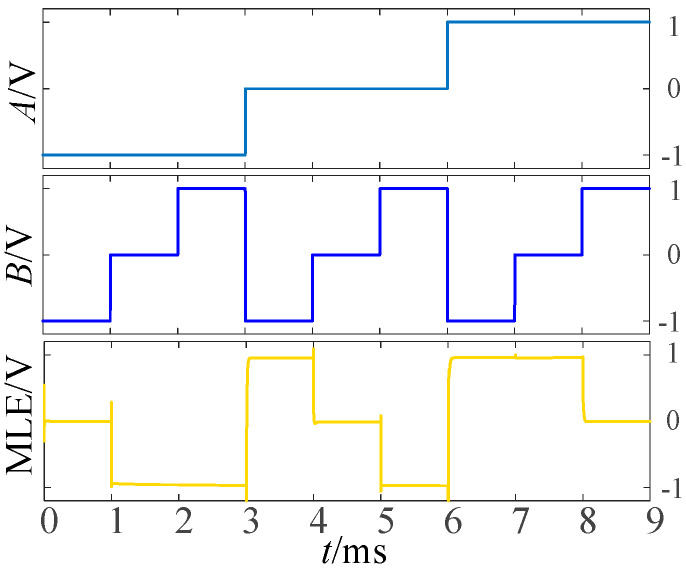
Simulation diagram of numeric comparator.

**Table 1 micromachines-14-01895-t001:** Balanced ternary univariate logic function truth table.

Input	Output												
*A*	*F* _1_	*F* _2_	*F* _3_	*F* _4_	*F* _5_	*F* _6_	*F* _7_	*F* _8_	*F* _9_	*F* _10_	*F* _11_	*F* _12_	*F* _13_
−1	−1	−1	−1	−1	−1	−1	−1	−1	−1	0	0	0	0
0	−1	−1	−1	0	0	0	1	1	1	−1	−1	−1	0
1	−1	0	1	−1	0	1	−1	0	1	−1	0	1	−1
Output													
*F* _14_	*F* _15_	*F* _16_	*F* _17_	*F* _18_	*F* _19_	*F* _20_	*F* _21_	*F* _22_	*F* _23_	*F* _24_	*F* _25_	*F* _26_	*F* _27_
0	0	0	0	0	1	1	1	1	1	1	1	1	1
0	0	1	1	1	−1	−1	−1	0	0	0	1	1	1
0	1	−1	0	1	−1	0	1	−1	0	1	−1	0	1

**Table 2 micromachines-14-01895-t002:** Structure diagram of three-state to two-state logic circuit and threshold voltage of MOS transistor.

Logic Function	*F* _4_	*F* _5_	*F* _9_
Circuit Structure	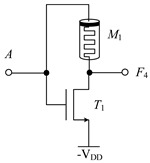	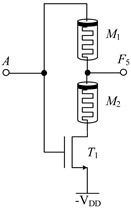	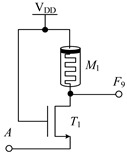
MOS Transistor Threshold Voltage	V_DD_ < *v*_th1_ ≤ 2V_DD_	V_DD_ < *v*_th1_ ≤ 2V_DD_	V_DD_ <*v*_th1_ ≤ 2V_DD_
Logic Function	*F*_10_, *F*_13_	*F* _18_	*F*_23_, *F*_26_
Circuit Structure	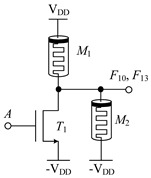	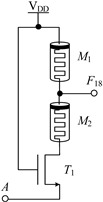	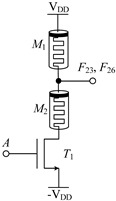
MOS Transistor Threshold Voltage	*F*_10_: 0V < *v*_th1_ ≤ V_DD_ *F*_13_: V_DD_ <*v*_th1_ ≤ 2V_DD_	V_DD_ <*v*_th1_ ≤ 2V_DD_	*F*_23_: 0V < *v*_th1_ ≤ V_DD_ *F*_26_: V_DD_ <*v*_th1_ ≤ 2V_DD_

**Table 3 micromachines-14-01895-t003:** The scheme of univariate three-state to two-state logic circuit designed via the cascade method.

Logic Function	*F* _2_	*F* _3_	*F* _7_	*F* _11_	*F* _15_	*F* _17_	*F* _21_	*F* _24_
Composition	*F*_26_ + *F*_4_	*F*_25_ + *F*_19_	*F*_4_ + *F*_9_	*F*_4_ + *F*_10_	*F*_25_ + *F*_26_	*F*_4_ + *F*_18_	*F*_4_ + *F*_19_	*F*_4_ + *F*_23_

**Table 4 micromachines-14-01895-t004:** Circuit structure diagram of up-spin logic function, down-spin logic function and threshold voltage of MOS transistor.

Logic Function	Up-Spin Logic Function, *F*_16_	Down-Spin Logic Function, *F*_20_
Circuit Structure	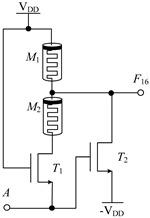	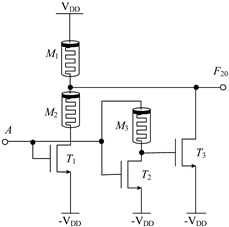
MOS Transistor Threshold Voltage	*T*_1_:*v*_th1_ > V_DD_ *T*_2_:*v*_th2_ ≤ 2V_DD_	*T*_1_:*v*_th1_ > V_DD_ *T*_2_:*v*_th2_ ≤ 2 V_DD_ *T*_3_:0V < *v*_th3_ ≤ V_DD_

**Table 5 micromachines-14-01895-t005:** Design scheme of univariate three-state to three-state logic circuit designed via cascade method.

Logic Function	*F* _8_	*F* _12_
Composition	*F*_20_ + *F*_22_	*F*_16_ + *F*_22_

**Table 6 micromachines-14-01895-t006:** Truth table of balanced ternary half adder, balanced ternary multiplier, and balanced ternary numerical comparator.

Input		Output			
		Half Adder		Multiplier	Numeric Comparator
*A*	*B*	SUM	CARRY	MUL	MLE
−1	−1	1	−1	1	0
−1	0	−1	0	0	−1
−1	1	0	0	−1	−1
0	−1	−1	0	0	1
0	0	0	0	0	0
0	1	1	0	0	−1
1	−1	0	0	−1	1
1	0	1	0	0	1
1	1	−1	1	1	0

**Table 7 micromachines-14-01895-t007:** Circuit structure of each application.

Half Adder	Multiplier	Numeric Comparator
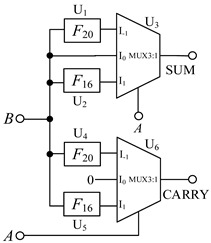	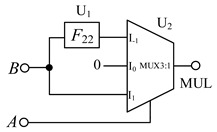	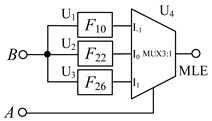

**Table 8 micromachines-14-01895-t008:** Comparison of the number of components of the circuits (here, T represents the number of transistors, M represents the number of memristors, and THA, MUL, and MLE are balanced ternary half adders, multipliers, and numerical comparators, respectively).

Method	Components		
	THA	MUL	MLE
Method in This Paper	46 (13T33M)	23 (7T16M)	29 (9T20M)
Multiplexer-Based Method in [31]	74 (10T64M)	38 (10T28M)	56 (10T46M)

**Table 9 micromachines-14-01895-t009:** Comparison of power consumption statistics of designed circuits.

Method	Avg. Power (uW)			Static Power (uW)			Dynamic Power (mW)		
	THA	MUL	MLE	THA	MUL	MLE	THA	MUL	MLE
Method in this paper	246.99	72.84	0.31	698 [−1&1]	193 [−1&1]	1.88 [0&−1]	3.61	4.56	1.44
Method in [27]	72.65	72.84	0.56	201 [−1&1]	181 [0&1]	1.51 [0&−1]	5.06	4.59	1.63

## Data Availability

The datasets generated and analyzed during the current study are available from the corresponding author on reasonable request.

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
