# Peer review of "Design and Application of Memristive Balanced Ternary Univariate Logic Circuit"

_micromachines, 2023, doi:10.3390/mi14101895_

Round 1
Reviewer 1 Report
In this manuscript, Wang et al. reported that the very interesting implementation of some common application-type combinational logic circuits of balanced ternary logic with memristor device, which can greatly enhance the efficiency of in-memory computing through multi-valued logic algorithms. The manuscript shows significant novelty in developing memristive logic circuits and is organized as well. I would like to recommend the manuscript being published on this journal after addressing some minor revision as following:
1. References provide an overview of the entire research work. It is recommended that the author include more references from the last three years.
2. The authors should also carefully check the reference format, e.g. reference[3], “Design of Ternary Logic and Arithmetic Circuits Using GNRFET” should be “Design of ternary logic and arithmetic circuits using GNRFET” instead
3. In the second paragraph of the introduction, lines 3-5, the author mentioned that memristor-based ternary logic provides advantages of non-volatility, nanoscale and compatibility with CMOS technology. The authors should add some state-of-the-art references or evidences to support your article?
4. There may be some grammatical and spelling mistakes in the manuscript, e.g. on page 5 line 4, “Therefore, only 5 kinds of three-state to three-state logic are effective and used in the circuit design.” should be “Only five types of three-state to three-state logic are effective and used in circuit design.” instead. On page 7, in equation (1), 'I-1' should be aligned with 'I0' and 'I1'. The authors are suggested to pay attention to English grammar and spelling so that the goals and results of the study are clear to the reader.
5. In Table 9, the static power consumption of the circuits designed in this paper is higher than that of the circuits designed in reference [27], with THA's static power consumption exceeding three times. It is recommended that the author provide an explanation in revised manuscript.
Minor editing of English language required
Reviewer 2 Report
The authors present a memristor-based design scheme for a balanced ternary digital logic circuit. To validate they proposed method LTSpice is used. The use of memristor for ternary circuits has been discussed for some time in literature, but the authors present a discussion about the consequences of their proposed new design and possible advantages. The author's manuscript proposal is very interesting from this point of view. However, I believe there are some possible points that should be clarified before the article is considered for publication.
The authors present simulation results but is not clear how the simulation was performed. Considering that memristor were used it would be important to explain how the memristor's state equations were defined, since there are many possible state equations for it. The authors could also discuss how different types of memristor would affect the designed circuit.
In the context of open science, the code or pseudo code could be presented, such the simulations could be verified and validated by others.
The more serious issue is related to reference 27. The authors compare the manuscript results with the results previously discussed in reference 27. However, the reference 27 is only a Arxiv manuscript and has not been published. Many details that should be in the present manuscript are pointed to reference 27. Considering that reference 27 was not published, I believe that the discussion should be in the other way around. The present manuscript should be used to discuss unpublished results. To avoid this confusion, the authors could compare their results with other already published works from other authors.
Based on all that, I believe the article requires revision before publication.
Round 2
Reviewer 2 Report
I am satisfied with authors comments, however I still believe that the memristor features and some details about its implementation should be better discussed in the text because there are many memristor types and they can behave differently.
Author Response
Thanks for your comments.
We have reply the similar questions in last respond, and have stated that all the memristors used in our simulation are broadly accessible Knowm memristors, we realize it in LTspice using the code provided by Knowm company. We have provided the reference of in last reply document too as below:
References
[1]Wang X Y, Dong C T, Zhou P F, et al. Low-variance memristor-based multi-level ternary combinational logic[J]. IEEE Transactions on Circuits and Systems I: Regular Papers, 2022, 69(6): 2423-2434.
[2]T. W. Molter and M. A. Nugent, “The generalized metastable switch memristor model,” in Proc. 15th Int. Workshop Cellular Nanosc. Netw. Appl. (CNNA). Frankfurt, Germany: VDE, 2016, pp. 1–2.
So we don't think we need to add any text on this issue.